# Crucial Regulatory Role of Organokines in Relation to Metabolic Changes in Non-Diabetic Obesity

**DOI:** 10.3390/metabo13020270

**Published:** 2023-02-14

**Authors:** Hajnalka Lőrincz, Sándor Somodi, Balázs Ratku, Mariann Harangi, György Paragh

**Affiliations:** 1Division of Metabolic Diseases, Department of Internal Medicine, Faculty of Medicine, University of Debrecen, H-4032 Debrecen, Hungary; 2Department of Emergency Medicine, Faculty of Medicine, University of Debrecen, H-4032 Debrecen, Hungary; 3Institute of Health Studies, Faculty of Health Sciences, University of Debrecen, H-4032 Debrecen, Hungary; 4Doctoral School of Health Sciences, University of Debrecen, H-4032 Debrecen, Hungary

**Keywords:** metabolically healthy obesity, non-diabetic obesity, adipokine, hepatokine, myokine, sarcopenic obesity, prediabetes, insulin resistance, cardiovascular risk

## Abstract

Obesity is characterized by an excessive accumulation of fat leading to a plethora of medical complications, including coronary artery disease, hypertension, type 2 diabetes mellitus or impaired glucose tolerance and dyslipidemia. Formerly, several physiological roles of organokines, including adipokines, hepatokines, myokines and gut hormones have been described in obesity, especially in the regulation of glucose and lipid metabolism, insulin sensitivity, oxidative stress, and low-grade inflammation. The canonical effect of these biologically active peptides and proteins may serve as an intermediate regulatory level that connects the central nervous system and the endocrine, autocrine, and paracrine actions of organs responsible for metabolic and inflammatory processes. Better understanding of the function of this delicately tuned network may provide an explanation for the wide range of obesity phenotypes with remarkable inter-individual differences regarding comorbidities and therapeutic responses. The aim of this review is to demonstrate the role of organokines in the lipid and glucose metabolism focusing on the obese non-diabetic subgroup. We also discuss the latest findings about sarcopenic obesity, which has recently become one of the most relevant metabolic disturbances in the aging population.

## 1. Introduction

The prevalence of obesity and associated comorbidities, i.e., insulin resistance, type 2 diabetes mellitus, accelerated atherosclerosis are extensively increasing [1]. These conditions are driven by several factors, including sedentary lifestyle, physical inactivity, fat and carbohydrate-rich food, genetic and environmental factors, as well as aging. Based on healthcare data, the worldwide prevalence of obesity has nearly tripled since 1975 with an estimated 135 million people being overweight or obese in Europe [2]. However, the ratio of overweight or obesity shows a high variability according to the different geographical region, such as 42.2% in the general adult population in the United States [3]; 44.8% among the Latin population in the United States [3]; 17.8% in the overall population in Dubai [4]; 21.6% in South African subjects [5] and 47.6% in Europe [6]. As a matter of fact, data from Central Eastern European region are more concerning than those from Western European countries; for instance, in the Hungarian population, the overweight (body mass index; BMI: ≥25 kg/m^2^) and obesity (BMI: ≥30 kg/m^2^) prevalence are approximately 35% and 20%, respectively [7]. Unfortunately, the estimated data are also very unfavorable, the World Obesity Atlas 2022 predicts that one billion people globally, including 1 in 5 women and 1 in 7 men, will be living with obesity by 2030 [8].

Obesity is characterized by excessive accumulation of fat leading to a plethora of medical complications, including coronary artery disease, hypertension, type 2 diabetes mellitus or impaired glucose tolerance and dyslipidemia. Enhanced atherogenesis, and therefore, premature atherosclerosis is also associated with the above-mentioned diseases and result in early cardiovascular complications and increased mortality. Due to its adverse effect on most of the cardiovascular risk factors, obesity increases the incidence of hypertension, coronary heart disease, heart failure and atrial fibrillation [9]. White adipose tissue, particularly visceral fat secreted adipokines, liver-expressed hepatokines and skeletal muscle-termed myokines may play a harmful or beneficial role in the crosstalk between the cells and may act as a “fine-tuner” of endocrine, autocrine and paracrine actions [10,11] (Figure 1).

Formerly, several physiological roles of these organokines have been described in obesity, especially in the regulation of glucose and lipid metabolism, insulin sensitivity, oxidative stress and low-grade inflammation [12]. However, it is important to note that obesity does not necessarily constitute the first step in the development of prediabetes and, subsequently, diabetes. Genetic background, ethnicity, post-translational modification and environmental effects, such as social globalization, urbanization, modern lifestyle, socioeconomic status are also causal factors in this process, as demonstrated on the upper part of Figure 1. Examining these bioactive peptides separately is insufficient, as they act together resulting in a complex network of actions in different tissues [13]. Obese subjects can demonstrate a varied spectrum of obesity-related complications [14]; however, a subset of obese individuals has a relatively normal metabolic homeostasis without manifesting carbohydrate disturbances. These subjects constitute the metabolically healthy obese (MHO) phenotype, and, despite data of previous findings, they have an increased risk of coronary heart disease compared to metabolically healthy normal-weight individuals [15]. Thus, these subjects cannot be considered healthy from all point of view and early changes in the crosstalk between organokines and lipid/glucose parameters can be detected. Non-diabetic obese (NDO) patients are characterized by obesity and certain metabolic disturbances including dyslipidemia, but their carbohydrate metabolism is within normal ranges. These phenotypes represent an intermediate status between healthy subjects and diabetic obese patients with altered organokine pattern, chronic inflammation and increased cardiovascular risk (Figure 2). In the last decade, our research group has investigated the potential role of bioactive peptides including a series of organokines regulating lipid and carbohydrate metabolism in NDO patients. In this review, we summarize the above-mentioned obesity-phenotypes and provide an overview of the organokines and their role in the lipid and glucose metabolism in non-diabetic obesity. We also discuss the latest findings about sarcopenic obesity, which has recently become one of the most relevant metabolic disturbances in the aging population.

## 2. Role of Novel Adipokines in Non-Diabetic Obesity

White adipose tissue, especially the visceral compartment is considered not only a simple energy reservoir, but also an active endocrine organ secreting a variety of biologically active polypeptides termed adipokines [16]. Several adipokines exist, of which leptin and adiponectin belong to the group of “classical adipokines” [17]; while later several “novel adipokines” have been identified, including plasminogen activator inhibitor-1 (PAI-1), chemerin, pigment epithelium-derived factor (PEDF), progranulin (PGRN), visfatin, omentin-1, vaspin, resistin, lipocalin-2 (NGAL) and meteorin-like/Metrln (IL-41) [18,19,20]. Here, we review the latest literature on novel pro- and anti-inflammatory adipokines and highlight their roles in non-diabetic obesity as well as in the obesity-related complications.

Chemerin is a 16 kDa adipokine secreted as an 18 kDa inactive pro-protein which can be activated by an extracellular serine protease cleavage of the C-terminal portion generating the active forms of chemerin, i.e., chem158K, chem157S and chem156F that are secreted into the bloodstream [21]. Previously, chemerin expression was found to be dramatically higher both in omental and subcutaneous adipose tissue in obesity [22]. Chemerin is closely associated with obesity-linked low-grade inflammation. Tumour necrosis factor-α treatment significantly enhanced the mRNA levels of chemerin in visceral adipocytes of obese patients suggesting that chemerin may act as a chemoattractant protein during inflammation [23]. Preliminary results showed that elevated chemerin may be associated with increased systolic blood pressure in obese children with overt arterial hypertension [24]. Interestingly, a study on hypertensive rats demonstrated that the chemerin stimulated smooth muscle cell proliferation and migration through activation of autophagy lead to vascular structural remodelling [25]. These and other investigations suggested a mechanistic influence of chemerin over blood pressure and hypertension [26]. In addition, in COVID-19 patients, elevated plasma chemerin was found to be a marker of disease severity and suggested to be an independent risk factor for the mortality [27]. Therefore, determination of serum chemerin could be an excellent disease marker in the daily clinical practice both in diseases with low-grade inflammation and in acute illnesses. On the other hand, chemerin showed positive associations with dyslipidemia, enhanced atherosclerosis and endothelial dysfunction. We previously demonstrated strong correlations between chemerin and lipoprotein subfractions in NDO subjects (mean BMI was above 40 kg/m^2^) [28]. There were positive correlations between large low-density lipoprotein (LDL), small-dense LDL, small high-density lipoprotein (HDL) subfractions and chemerin, while negative correlations were found between mean LDL size, large HDL, intermediate HDL subfractions and chemerin. In line with former findings [29], we hypothesized that chemerin might modify the composition of LDL in the atherosclerotic lesion due to its role in low-grade inflammation and via its paracrine effects. These processes might lead to the development of the small-sized, more atherogenic LDL particles promoting the increased risk of atherosclerosis. In this study, multivariate analysis showed that C-reactive protein (CRP) and small HDL but not BMI were independent predictors of chemerin. Due to the obesity-related inflammation, compositional alterations might develop within the HDL particle, especially in the core lipid moiety, including reduction in the free cholesterol and cholesteryl-ester content, which may result in decreased HDL size and density [30]. This phenomenon can partially explain our observation.

Human paraoxonase-1 (PON1) is a HDL-linked esterase, which has marked antioxidant properties and the ability to counteract atherosclerotic processes by preventing LDL from oxidative modification [31]. In one of our studies, we found significant negative correlations between PON1 arylesterase activity, adiponectin and chemerin, while chemerin correlated positively with leptin in NDO subjects [32]. These results indicate that chemerin may act as a potential modulator of inflammation, and it can be associated with enhanced atherosclerosis and impaired antioxidant status in obese subjects even without causing insulin resistance (Figure 3).

Plasminogen activator inhibitor-1 (PAI-1) is an adipokine with potent fibrinolytic activity. Elevated level of circulating PAI-1 is well-documented in obesity and type 2 diabetes [33]. Plasma PAI-1 is involved in lipid metabolism and positively correlated with triglyceride levels, while negatively correlated with the concentrations of HDL-cholesterol (HDL-C) in healthy young adults [34] However, the exact link between HDL subpopulations and PAI-1 is not fully clarified. Using sequential ultracentrifugation, small-sized HDL, but not large-sized HDL subfraction stimulated PAI-1 release in the murine 3T3 adipocyte cell line [35]. Previously, our research group investigated the concentration of PAI-1 in NDO patients and demonstrated significantly higher PAI-1 levels in the obese group compared to lean controls [36]. There were strong significant negative correlations between PAI-1 and the levels of large and intermediate HDL subfractions, as well as mean LDL size. Based on a multiple regression analysis, waist circumference and intermediate HDL subfraction were predictors of plasma PAI-1 concentration. Our findings highlight the relationship between PAI-1 and lipoprotein metabolism in obesity.

A common 4G/5G-675 single guanine insertion/deletion polymorphism in the promoter region of the PAI-1 gene is of functional importance in regulating PAI-1 expression [37]. The presence of the 4G allele does not influence plasma PAI-1 levels under physiological state; however, it may exacerbate the effect of several factors on PAI-1 level during certain pathological conditions [38,39]. Additionally, previous studies suggested that very-low density lipoprotein (VLDL) was capable of increasing the PAI-1 level via a VLDL response element localized in the promoter region of the PAI-1 gene, mediating VLDL-induced PAI-1 transcription in endothelial cells [40,41]. In a recent pilot study [42], the genotype distribution of PAI-1 4G/5G polymorphism did not significantly differ in NDO individuals compared to normal-weight controls. There were correlations between PAI-1 levels and the components of metabolic syndrome suggesting a closer link between PAI-1 and lipid as well as carbohydrate metabolism in subjects with 5G/5G genotype. However, further studies on larger number of individuals are needed to clarify this phenomenon.

Lipocalin-2 (LCN-2) or neutrophil gelatinase-associated lipocalin (NGAL) is a member of the lipocalin superfamily and is highly expressed by different types of tissues including omental adipose tissue, liver, kidney, spleen, heart, lung, bone marrow and skin [43]. LCN-2 is a glycoprotein consisting of 198 amino acids in humans. The human LCN-2 protein structure shows only approx. 60% similarities with rodents but LCN-2 domain architecture and signaling peptide is highly uniform [43]. These facts could help us to understand the results found in rodents. Serum and urinary levels of LCN-2 are associated with the markers of obesity, insulin resistance and diabetes both in adults and adolescents [44,45,46]. It was reported that circulating levels of LCN-2 was markedly increased in subjects that had impaired fasting glucose and glucose tolerance compared to normal subjects. This result suggests that LCN-2 upregulation is associated with a higher risk for impaired glucose regulation and type 2 diabetes [45]. In addition, urinary LCN-2 may be a potential biomarker for early renal injury in insulin resistant obese non-diabetic children [47], in obese prepubertal children [48], as well as in H-type hypertension (homocysteine ≥ 10 μmol/L) [49]. Interestingly, LCN-2 levels increased after high-intensity interval training in obese men [50] and after a high-intensive vertical run [51]. It has also been demonstrated that LCN-2 reduces myogenesis, suggesting that LCN-2 may negatively affect muscle physiology when upregulated following high-intensity exercise in mice.

Pigment epithelium derived factor (PEDF) is originally described as a member of the serine protease inhibitor family and is secreted by the human retinal pigment epithelial cells and the visceral adipose tissue [52,53]. This glycoprotein has significant antiangiogenic properties by direct effects of endothelial cells via Akt/MAPK, Wnt/β-catenin and NFAT/c-FLIP pathways [54] and plays a role in lipid metabolism by the binding of adipose triglyceride lipase [55]. On the other hand, higher circulating PEDF was also associated with adiposity, with the degree of insulin resistance, as well as with obesity [56,57]. In a Czech study, significantly higher PEDF was found in type 2 diabetic patients with metabolic syndrome; and the von Willebrand factor was independently associated with PEDF in type 2 diabetic patients without metabolic syndrome indicating the potential angio-protective role of PEDF in diabetes [58]. One may hypothesize that under obesity-induced pathological conditions PEDF expression could be upregulated as a compensatory mechanism. This hypothesis is supported by data on Otsuka Long-Evans Tokushima Fatty (OLETF) rats, an animal model of type 2 diabetes with obesity, hence long-term administration of recombinant PEDF significantly improved body weight, metabolic parameters and insulin resistance [59]. Data of PEDF concentration on severely obese subjects without carbohydrate disturbances are completely lacking; therefore, further clinical studies are needed to clarify the role of PEDF in obesity.

Progranulin (PGRN) is originally identified as a growth factor and expressed by a wide range of cell types including epithelial cells, fibroblasts and adipocytes [60]. In mice, PGRN binds directly to tumor necrosis factor receptor (TNFR) inhibiting neutrophil activation and disturbing the interaction between TNFα and TNFR [61]. Thus, PGRN may inhibit TNFα-induced activation of the nuclear factor-kappa B (NF-κB) and mitogen-activated protein kinase (MAPK) signaling pathway [62]. A growing number of human studies have demonstrated the possible role of PGRN in insulin resistance and obesity, but data are highly controversial. Generally, hyperprogranulinemia has been reported to be linked to adiposity, metabolic parameters, insulin resistance and inflammatory markers both in adult [63] and childhood obesity [64]; however, in other studies PGRN has not been found to be associated with insulin sensitivity and beta-cell function [65]. In obese healthy men, significantly improved body composition and insulin resistance and decreased serum PGRN levels were detected after 8-week circuit resistance training [66]. Circulating PGRN/TNFα ratio was found to be an independent predictor of systolic blood pressure in overweight hypertensive adults [67]. In another interventional study, 6-month alpha-lipoic acid treatment significantly increased circulating PGRN levels; in addition, the improvement of current perception threshold correlated negatively with PGRN after treatment in obese type 2 diabetic patients [68]. Interestingly, Brock et al. [69] investigated serum PGRN concentrations as well as subcutaneous and visceral adipose tissue PGRN expression in morbid obese patients undergoing bariatric surgery. PGRN expression in the subcutaneous adipose tissue was higher than in the visceral adipose tissue; and importantly, serum PGRN concentrations were independent from type of adipose tissue expression.

Visceral adipose tissue-derived serpin protease inhibitor (vaspin or SERPINA12) belongs to the serpin superfamily and is firstly described in OLETF rats in 2005 [70]. In this study, after insulin treatment vaspin was increased in the bloodstream at 6 weeks and highly expressed in the adipose tissue at 30 weeks. Additionally, both circulating and the expression levels of vaspin were normalized after insulin treatment in OLETF rats. In addition, administration of vaspin to diet induced obese mice significantly improved glucose tolerance and insulin sensitivity [70]. These findings suggested that vaspin may serve as an insulin-sensitizing and anti-inflammatory agent via a compensatory mechanism, which is activated in response to the insulin resistance [71]. Vaspin may be involved in hypertension by a complex interaction with other adipokines, cytokines and oxidative stress pathways in overweight hypertensive subjects [72].

In some human studies including meta-analyses, circulating vaspin concentration was found to be generally higher in obese individuals compared to non-obese controls [73,74] but other studies did not confirm this observation. Indeed, a recent study showed that vaspin levels were significantly lower and showed positive correlation with HDL-C in overweight and obese children with NAFLD [75]. In Saudi patients with T2DM, serum vaspin was significantly lower than the controls [76]. These results may highlight the beneficial effects of vaspin on the atherogenic lipid profile and suggest the usefulness of vaspin as a biomarker in obesity and diabetes.

Omentin-1 is an anti-inflammatory and insulin sensitizing adipokine and mainly expressed by stromal vascular cells in the omental visceral tissue [77]. Omentin-1 increased Akt phosphorylation in the insulin signaling pathway in the absence and presence of insulin [77]. In human studies, serum omentin-1 concentrations were lower in obese patients and inversely correlated with the markers of obesity [78], metabolic syndrome [79] and diabetic complications [80]. Low omentin-1 level has been suggested to be a reliable marker to predict the development of MetS in patients with hypertension [81]. Indeed, omentin-1 was lower in both pre- and postpubertal obese children with metabolic syndrome compared to obese children without metabolic syndrome. In this study, waist circumference correlated negatively with omentin-1, while triglyceride and HDL-C showed positive correlations with omentin-1 among girls as well as in the postpubertal subgroup [82]. In addition, serum omentin-1 significantly increased after short-term and long-term lifestyle interventions [83]. Since atorvastatin administration increased the circulating omentin-1 levels, omentin-1 may be involved in the lipid profile regulation [84].

Meteorin-like (Metrnl)/interleukin-41 (IL-41) is a poorly characterized white adipose tissue-secreted small protein (∼28 kDa) and is associated with thermogenesis in the beige/brown adipocytes [85]. High levels of Metrnl were found to be stimulated by IL-4 expression through eosinophil dependent signaling, which in turn stimulates the macrophages in the adipose tissue that activates the thermogenic gene and anti-inflammatory programs in adipose tissue [85]. In addition, Metrnl improved lipopolysaccharide-induced inflammatory responses via MAPK or PPARδ-mediated signaling pathways [86]. Human data on Metrnl are largely controversial. Low circulating Metrnl level is inversely associated with glucose levels and insulin resistance in type 2 diabetes [87], as well as renal function and diabetic nephropathy [88]. In contrast, others found increased Metrnl levels in newly diagnosed diabetic patients with positive associations between Metrnl and lipid and glucose profile and insulin resistance [89].

An increasing number of publications describe the pleiotropic effects of novel adipokines on metabolic homeostasis. In this section, we summarized the most relevant papers related to obesity and related complications especially in non-diabetic obese individuals; however, we did not aim to present the entire literature. In some cases, data are lacking or highly controversial about the exact physiological role of adipokines in the metabolic processes but measurement of circulating adipokines might be a novel therapeutic approach to map the status of obese patients.

## 3. Role of Novel Hepatokines and Gastrointestinal Hormones in Non-Diabetic Obesity

Afamin is a vitamin E and D binding glycoprotein, and beside albumin, vitamin D binding protein and α-fetoprotein, is the fourth member of the albumin family [90]. It is mainly expressed by hepatocytes and based on the contradictory results of previous papers, the exact (patho)physiological functions of afamin are largely inconsistent. Afamin may serve as a tumor marker for different types of malignancies, i.e., ovarian, breast and thyroid cancers [91,92,93]; additionally, it may be involved in the bone metabolism [94] as well as in the pathogenesis of Alzheimer’s disease [95]. Higher afamin levels are significantly associated with the markers of obesity and the obesity-linked disorders such as metabolic syndrome, type 2 diabetes, and gestational diabetes (Figure 4) [96,97]. In a large multicenter population-based study, circulating afamin was associated independently with prevalent and incident type 2 diabetes (OR 1.30 [95% CI 1.23–1.38]; *p* < 0.001) indicating that afamin can be a promising novel biomarker to identify subjects with a high risk of type 2 diabetes [98]. Proteomic profiling analyses showed that serum protein concentrations of afamin are correlated with the instability of atherosclerotic plaques in men with coronary atherosclerosis [99]. In addition, circulating afamin are increased in patients with NAFLD and after adjusting potentially confounding parameters, independently predicted the development of NAFLD [100]. A very recent study demonstrated that serum afamin is markedly elevated in NDO individuals and did not correlate with α- and γ- tocopherol levels [101]. However, there were positive correlations between BMI, fasting glucose, triglyceride, oxidized LDL and afamin in this study. Bidirectional correlations were seen between afamin and HDL subfractions (measured by Lipoprint^®^); large-sized HDL subfractions correlated negatively, while small-sized HDL subfractions correlated positively with afamin. Since afamin was predicted by small HDL—beside waist circumference and HbA1c—in a multiple regression model, this organokine might play an important role in the regulation of lipid metabolism, especially in the HDL function. However, it is not clear whether increased afamin level in obesity contribute to the development of insulin resistance or is simply an unrelated consequence of obesity. Further studies are needed to clarify the role of afamin in obesity and the development of insulin resistance.

Retinol-binding protein 4 (RBP4) is mainly synthetized and secreted by the liver as well as the adipose tissue [102], and transfers the active form of vitamin A, namely retinol, with transthyretin from the liver to the extrahepatic tissues in the circulation [103]. Although several studies have investigated the function of RBP4, data from physiological and pathological levels of RBP4 are still inconsistent. Previously, a U-shaped association was described between circulating RBP4 and the risk of incident of type 2 diabetes in middle-aged Chinese prediabetic patients [104] and in women with high cardiometabolic risk [105]. In addition, other papers published higher serum RBP4 levels in obesity, insulin resistance, type 2 diabetes, metabolic syndrome and in patients with cardiovascular disease [103,106,107]. However, a population-based study on Polish participants did not confirm these results and the authors described no significant differences between obese and normal-weight controls [108].

In the NDO population, unexpectedly lower serum RBP4 was found compared to healthy lean participants, and the levels of RBP4 correlated positively with BMI, waist circumference and fasting glucose in overall subjects, as well as with C-peptide in the NDO subgroup [109]. It is important to mention that the type of laboratory assay for the RBP4 measurement (Western blotting, enzyme linked immunosorbent assay, enzyme immunoassay) might be crucial in the determination of circulating RBP4 concentration [103]. Beside the previously mentioned disorders, RBP4 and retinoids are also associated with pro-atherogenic lipid profile, especially with hypertriglyceridemia in subjects with high type 2 diabetes risk [110]. Our research group found strong positive correlations between ApoAI, HDL-C, intermediate HDL subfraction, small HDL subfraction and VLDL subfraction (subfractions were measured by Lipoprint^®^) in NDO and lean subjects [109]. Triglyceride levels were marginally correlated with RBP4. Multiple regression analysis showed that VLDL subfraction is an independent predictor of RBP4 demonstrating the potential involvement of RBP4 in the lipid metabolism in obesity. A previous study partly supported these findings, since circulating RBP4 positively correlated with triglyceride levels but not with BMI, fat mass and insulin resistance in healthy obese and non-obese individuals [108].

Besides RBP4, Fetuin-A or α-2-Heremans-Schmid-glycoprotein (α2-HS-glycoprotein, AHSG) is also a multifactorial organokine mostly produced by the liver and partially by the visceral adipose tissue [111]. In humans and animals, fetuin-A is originally described as an essential developmental factor in the fetal circulation [111] and it serves as a potent inhibitor of vascular calcification in endothelial cells [112]. Fetuin-A level is considered a biomarker for obesity [113], dyslipidemia [114], subclinical atherosclerosis [115] and thyroid diseases [116]. A recent meta-analysis suggests that fetuin-A may prove to be an important indicator for the components of metabolic syndrome [117]. In addition, high fetuin-A level played a role in the development of insulin resistance through its effect on the endogenous inhibitor of insulin receptor tyrosine kinase [118]. Some reports indicated elevated levels of fetuin-A in obese individuals [119] and in patients with type 2 diabetes [118]; while others could not confirmed these results [109,120]. Besides non-genetic influences, AHSG single nucleotide polymorphisms (SNPs) were found to be susceptible for obesity and related early-onset comorbidities [121], while other SNPs were associated with cardiovascular outcomes partly in patients with diabetes [122]. Proteomic analyses described an extended list of HDL-associated proteins, and these experimental data demonstrated that RBP4 and fetuin-A could associate with HDL besides the widely-known HDL-bound structural and enzymatic proteins [123]. However, clinical results are contradictory about the link between fetuin-A and HDL. In a middle-aged NDO cohort, fetuin-A correlated positively with HDL-C, apoAI and large-sized HDL subfractions [109]. Furthermore, fetuin-A correlated positively with RBP4 in this study. These results corroborated by other publications [124,125], since during unfavorable oxidative milieu RBP4 and fetuin-A may be significantly enriched in HDL particles in the circulation leading to impaired HDL proteome composition via N-glycosylation and sialylation.

Fibroblast growth factor 21 (FGF21) is originally described as a liver-expressed organokine, but it is also secreted by the pancreas and skeletal muscle [126]. FGF21 has several beneficial effects on obesity and obesity-related disorders. FGF21 enhances glucose uptake and oxidation in an insulin-independent manner by inducing the expression of glucose transporter-1 in adipocytes and skeletal myocytes [127]. In addition, FGF21 may serve as an important regulator of thermogenesis, lipolysis and insulin-sensitization in rodents [128,129,130]. This hepatokine acts through a cell surface receptor, FGF receptor (FGFR) with their transmembrane co-receptor named β-Klotho activating FGF21 to enhance intracellular signaling pathways, which eventually leads to metabolic effects [131]. Since FGFRs are widely expressed in the white and brown adipose tissue, as well as in the central nervous system, growing evidence are available about the pleiotropic effects of FGF21 [132]. Serum FGF21 and vWF levels were increased in elderly patients with hypertension and associated with carotid atherosclerosis [133]. Paradoxically increased serum FGF21 level was found in patients with obesity and type 2 diabetes indicating a possible compensatory response to metabolic alterations [134]. Interventional studies showed that physical activity significantly improves circulating FGF21. For instance, in obese type 2 diabetic patients with peripheral neuropathy (BMI: 31.6 ± 3.9 kg/m^2^), 6 weeks of moderately intensive physical activity increased serum FGF21 [135]. In this study, the change in FGF21 negatively correlated with changes in BMI, improvement of current perception threshold value and TNFα levels, while there was a positive correlation between change of FGF21 and adiponectin. Other authors also published that high-intensity interval training had a favorable effect on FGF21 and irisin levels in overweight and obese men [136].

Obestatin is a little-known anorexigenic gut hormone consisting of 26 aminoacids and is derived from the same 117-residue preproghrelin in the C-terminal region [137]. The aminoacid sequence of obestatin is highly conservative among both primates and mammalians [138]. It seems like obestatin is related to insulin resistance, obesity and type 2 diabetes mellitus both in overweight and obese prepubertal children and adults [139,140]; however, the precise effects of obestatin are largely unknown in humans. In vitro studies revealed that obestatin is secreted in pancreatic islets and it has been reported to stimulate insulin secretion in the absence and presence of glucose [141]. Obestatin promoted β-cell survival, mass growth and differentiation, as well as indicated gene expression is associated with insulin production [142]. In addition, obestatin is involved in the regulation of lipid metabolism but the data are not fully explained. Studies on rats and mice demonstrated that continuous obestatin treatment reduced serum triglyceride levels, but cholesterol levels were unchanged [143]. In addition, others showed that during obestatin treatment the expression of cholesterol transporter ATP-binding cassette A1 (ABCA1) is decreased in bovine visceral adipose tissue suggesting changes in cholesterol transport [144].

Szentpéteri et al. described lower circulating obestatin in NDO patients compared to lean, healthy ones [145]. Negative correlations were found between BMI, glucose, insulin and HbA1c, VLDL subfraction and obestatin, while HDL-C, ApoAI and mean LDL size correlated positively with obestatin highlighting the possible favorable role of obestatin in morbid obesity. Negative correlation between VLDL subfraction and obestatin may be explained by the previously described findings, in which insulin resistance and the higher level of serum glucose resulted in increased hepatic free fatty acid production leading to elevated VLDL level [146]. The measurement of obestatin level may contribute to better understand the interplay between gut hormone secretion and metabolic alterations in obesity.

Ghrelin is an orexigenic gut hormone which is secreted by P/D1-type cells in the stomach and duodenum. Similar to obestatin, ghrelin is formed from the 117-amino acid preproghrelin through the post-translational cleavage leading to 28 amino acid inactive (unacylated) proghrelin [147]. Proghrelin is acylated by ghrelin O-acyltransferase (GOAT) at the third serine residue to produce active ghrelin [148]. This active form of ghrelin can bind to growth hormone secretalogue receptor type 1a (GHSR1a) regulating energy expenditure. Ghrelin increases appetite, stimulates gastric motility and hepatic glucose production, and reduces insulin secretion [147]. Both human and murine studies showed [149,150] that peripheral ghrelin administration stimulates food intake. Ghrelin also induces growth hormone and glucocorticoid secretion [151]. Previous studies showed that circulating ghrelin levels are decreased in human obesity compared to lean subjects [152]. Furthermore, the postprandial ghrelin levels are markedly decreased in obese individuals regardless of their gender [152]. Since the first description of GOAT in 2010, GOAT inhibitors have become a promising therapeutic strategy in the treatment of obesity and diabetes via their ability to increase insulin response and reduce ghrelin activity [153]. Many of these inhibitors have been disclosed by pharmaceutical patterns and clinical studies are ongoing in this field. However, some methodological limitations should be mentioned. The main challenge in the investigation of ghrelin is to measure the active form of ghrelin concentration which is normally only 10% in the bloodstream [154]. It was also reported that the active ghrelin is unstable at room temperature in the sample, and it seems the time of the sample collection as well as the storage of the samples are highly essential and need further standardization [155]. Moreover, the major number of enzymes linked immunoassays (ELISAs) are validated for the determination of total ghrelin level and do not differentiate the two forms of ghrelin. Besides ELISAs, a specific radioimmunoassay could be another way to determine ghrelin concentration [154]. The methodological differences may contribute to the non-comparable data in the literature.

## 4. Role of Myokines in Non-Diabetic Obesity

The skeletal muscle is the largest endocrine organ of non-obese individuals. Skeletal muscle cells secrete above six hundred peptides termed myokines, which have an important role in the regulation of muscle mass and function. They act as a mediator between skeletal muscle and other organs, such as the liver and the adipose tissue, and influence the glucose, lipid and amino acid metabolism and energy homeostasis. Myokine signaling pathways influence the proliferation and differentiation of muscle cells, the mitochondrial function, inflammatory response, fat browning and metabolic homeostasis [156].

After the fifth decade the body composition changes with aging. After this age the bone density decreases by about 1–1.5 percent per year, muscle mass by about 1.5–2 percent per year, and the muscle strength by about 2.5–3 percent per year [157]. Parallel with this process, the probability of obesity is increasing with age, which leads to higher body fat content [158]. Sarcopenia is defined as the progressive and generalized loss of skeletal muscle mass and strength, which increases the risk of adverse outcome and causes higher mortality [159]. There is no widely accepted criteria for sarcopenia, but all of them based on two or three of the following parameters: low muscle mass, low muscle strength and low physical performance [160]. The prevalence of sarcopenia is higher in women compared to men with all of definitions [161]. For measurement of muscle mass, dual energy X-ray absorptiometry or bioimpedance analysis can be used in the clinical practice. The muscle strength can be determined by handgrip strength, while the physical performance can be measured by “Short Physical Performance Battey” (SPPB), usual gait speed, walk test and stair climb power test [159].

Sarcopenic obesity means the co-existence of low muscle mass and strength and the high body fat content. A recent metanalysis showed that for the measurement of sarcopenia a lot of methods were used worldwide. Some studies used height or weight adjusted appendicular skeletal muscle (ASM), while others applied combined methods measuring the poor muscle mass and either poor strength or function for diagnosis of sarcopenia. For the diagnosis of obesity body fat percentage (BF%), BMI and waist circumference are available [162]. Sarcopenic obesity is mostly related to aging, which is accompanied by loss of muscle mass and parallel increase in fat mass. The sedentary lifestyle, the progressive decrease in physical activity and protein intake lead to reduction in lean body mass, which decrease the resting metabolic rate [163]. The low-grade inflammation in obesity and the intramuscular fat deposition leads to mitochondrial dysfunction and unfavorable myokine profile [164]. The mitochondrial β-oxidation is impaired, which causes lipid peroxidation, accumulation of reactive oxygen metabolites (ROMs) resulting in insulin resistance (IR), oxidative stress and lipotoxicity within the muscle cells [164]. The age-related changes of sex-specific hormones and growth hormone play an important role in sarcopenic obesity. Testosterone stimulates protein synthesis and increases muscle mass. In obesity, the testosterone level decreases because of increased aromatase activity, which contribute to sarcopenic obesity [165,166]. In menopause, an increased level of gonadotropins and androgens as well as a decreased level of estrogen lead to the deposition of visceral adipose tissue and to decreased fat free mass [167]. Growth hormone (GH) declines also with both aging and obesity. Low GH downregulates by decreased IGF-1 the PI3K-AKT/PKB-mTOR pathway, which induces the muscle protein synthesis [168]. According to the above mechanisms, decreased production of sex hormones and GH with aging aggravates the process of sarcopenia, which is accelerated in obesity. According to a recent meta-analysis the prevalence of sarcopenic obesity in adults aged ≥50 years is between 7% and 13% in different regions, the highest value was detected in Europe and in South America. No gender difference was reported. Based on the data of 34 studies, obesity decreased the risk of sarcopenia by 34% compared to normal weight population. Sarcopenic obesity was associated with elevated risk of all-cause mortality (HR 1.51 [1.14–2.02]; *p* < 0.001), while obese elderly patients had a similar risk of all-cause mortality, than that of healthy controls [162]. Sarcopenic obesity was associated significantly with CVD events (heart disease, stroke), metabolic disorders (metabolic syndrome, diabetes mellitus), cognitive impairment, lung diseases, physical disability and arthritis [162]. These findings demonstrate the complex interactions between obesity and sarcopenia with multiple—positive and negative—factors implicated in the maintenance of fat and muscle mass.

Irisin is an exercise-induced myokine, which is the cleavage product of fibronectin type-III domain-containing protein 5 (FNDC5) regulated by peroxisome proliferator-activated receptor-g co-activator 1a (PGC-1a) [169,170]. The FNDC5 expresses 200 times higher in muscle, than in fat cells [171]. Circulating irisin level is increased in individuals doing intensive physical activity and progressively reduced in less active patients and in the case of a sedentary lifestyle [172]. The cold weather and the leptin also increase the irisin level [173,174]. Animal models showed that exogenous FNDC5 induces UCP1 expression in subcutaneous white adipocytes, while FNDC5-treated adipocytes showed a brown fat-like phenotype with an elevated UCP1 expression [175]. FNDC5 overexpression in the liver and elevated circulating irisin level prevented diet-induced weight gain, metabolic disturbances, and stimulates oxygen consumption [176]. Irisin increases the glucose uptake by skeletal muscle cells, facilitates the hepatic glucose and lipid metabolism and decreases the hyperglycemia and hyperlipidemia [177]. It acts as an insulin sensitizing hormone, improves the hepatic glucose metabolism by reducing endoplasmic reticulum stress and promotes pancreatic β-cell function and survival [178]. According to these effects, irisin can reduce the risk of developing type-2 diabetes mellitus. Irisin stimulates glucose uptake by calcium/ROS and P38 AMPK mediated AMPK pathway involving the translocation of p38 MAPK-GLUT4 to the plasma membrane [179,180]. These results shows that irisin has an insulin sensitizing effect with consequences in treating diabetes mellitus [181].

Human studies showed conflicting results regarding the plasma level of irisin in diabetic patients. Surprisingly, lower levels of irisin were found in patients with known diabetes compared to new onset diabetes [182], or in non-diabetic subjects [183]. In another study, irisin is inversely correlated with insulin sensitivity [184]. In some human studies, irisin has been positively associated with the risk of metabolic syndrome, cardiometabolic disturbances, and cardiovascular disease [185]. A number of studies have examined the link between circulating irisin and adiposity and obesity in humans with controversial results [186]. Furthermore, irisin levels were reduced significantly following weight loss due to bariatric surgery, which was the consequence of the lower fat-free mass and decreased FNDC5 mRNA expression in the skeletal muscle [187]. Irisin levels were decreased in overweight/obese children with metabolic syndrome, therefore, irisin can be used as a biomarker for metabolic syndrome in prepubertal children [186]. In adipose tissue, it induces the phenotype switch of macrophages form M1 to M2 form, which has an anti-inflammatory effect [188] and stimulates the browning process by activating UCP1 expression. Dietary factors also influence the irisin level. Supplementation of diet with omega-3 fatty acids (EPA and DHA) elevated serum irisin level significantly [189]. Significant association can be found between circulating irisin and favorable lipid profile in the general population suggesting that increased irisin concentration is associated with low risk for chronic non-communicable diseases [190]. Despite these results, the associations of plasma irisin concentration with body composition, physical activity and dietetic interventions are controversial and still not completely understood.

Myostatin, also referred to as growth and differentiation factor 8 (GDF8) belongs to TGF-β superfamily and highly expressed in skeletal muscle. Myostatin is synthetized as a precursor protein and requires release from the propeptide to become biologically active form. Myostatin binds to the active type IIB receptor (ActRIIB) and forms a heterodimer with activin-like kinase 4 (ALK4) or ALK5, which in turn activates Smad2 and Smad3, then forms a complex with Smad4, which is transferred to the nucleus [191]. This complex influences transcription factors such as myocyte-specific enhancer factor (MEF2) and myoblast-determining protein (MyoD), which inhibits myoblast proliferation and differentiation. Myostatin also inhibits the Akt/mTOR pathway and therefore suppresses skeletal protein synthesis and acts also through FOXO-1 and accelerates skeletal muscle atrophy. Myostatin is also a strong negative regulator for satellite cell activation [192]. According to the above mechanisms, myostatin is a negative regulator of skeletal muscle growth. Myostatin level is influenced by physical activity, aging, NF-kB and TNF-α. Serum myostatin level increases with age in the general population and is inversely related to skeletal muscle mass in human studies [193]. In human sarcopenia myostatin mRNA and protein levels were elevated by 2 and 1.4-fold; therefore, it is a potential candidate influencing the anabolic perturbation in sarcopenic obesity [194]. The overexpression of myostatin has been shown to induce strong TNF-α expression [193], while the deletion of functional mutations in myostatin causes skeletal muscle hyperplasia and hypertrophy [191]. After 12 weeks of resistance and endurance training the concentration of serum myostatin was reduced, while the levels of irisin and follistatin increased simultaneously in obese people [195]. According to the above data, myostatin levels increase with age and persistent inflammation, and decrease with exercise; therefore, exercise can be used as an intervention to prevent sarcopenic obesity.

Follistatin was initially identified as a component of the follicular fluid that binds activins and neutralizes their bioactivity. Follistatin also binds with lower affinity to several other members of the TGF-β superfamily including myostatin and bone morphogenic proteins (BMPs). Two variants of follistatin are generated by alternate splicing at the C terminus of the common precursor gene, namely follistatin 315 and follistatin 288 which is lacked the carboxyl-terminal region of follistatin 315 [196]. Follistatin and myostatin interact directly with a high affinity. The myostatin-induced decrease in the expression levels of key myogenic proteins Pax3 and MyoD was significantly blocked in the presence of follistatin; therefore, follistatin is a positive regulator of muscle development. Follistatin-induced muscle hypertrophy was associated with inhibition of both myostatin and activin A and induction of satellite cell proliferation [197]. Transgenic expression of follistatin in mdx mice, a model for Duchenne muscular dystrophy, showed amelioration of dystrophic pathology and an increase in skeletal muscle mass [198]. Interestingly, in a gene therapy trial, Mendell et al. demonstrated beneficial effects of follistatin in direct delivery into intramuscular quadriceps in patients suffering from Becker Muscular Dystrophy without any apparent side effects [199]. Follistatin was identified as a direct target of the testosterone effect during its promyogenic action in both mouse models [200] and cell culture studies [201]. Follistatin significantly antagonized the TGF-β-induced inhibition of phosphorylation of Smad2/3 and MHCII expression in satellite cells [201]. Follistatin has a central role in promoting muscle mass and function, and studies approved the potential therapeutic use for the treatment of muscle wasting in cachexic conditions. In addition, follistatin has an important role in beige and brown adipose tissue differentiation from precursor cells by activation of p38MAPK and ERK1/2 signaling pathway [202]. On the basis of these results, follistatin is an excellent candidate for therapeutic intervention of obesity.

Here, we discussed the latest findings about sarcopenic obesity and some of related myokines; however, we did not aim to present the entire literature, since there are more than six-hundred myokines mentioned by former studies. On Table 1, we summarized the supposed pathophysiological role of organokines in the relation of obesity and obesity-related disturbances.

## 5. Treatment and Future Directions

A growing amount of evidence suggest that MHO phenotype is more susceptible for cardiometabolic disturbances and has worse all-cause mortality than we have thought before. Furthermore, a very recent study suggested that metabolically unhealthy phenotype was associated with an approximately 75% increased risk for cardiovascular disease compared to MHO [204]. However, when using class I obesity as reference, class III obesity was markedly associated with cardiovascular complications in the MHO group, also indicating both severity of obesity and metabolic status showed an increased risk for the incident of cardiovascular disease [204]. The situation is complicated by the fact that the definition of MHO varies considerably in the literature; over thirty distinct criteria for defining MHO are described worldwide [205]. Consequently, the prevalence of MHO phenotype also differs, ranging from 2.2 to 11.9% in the general population according to the definition of MHO [15,206]. It is also clear that only BMI-based classification as well as determination of major routine laboratory parameters, such as fasting glucose, insulin or lipid measurement are insufficient to assess the health status of obese patients; more complex and detailed evaluation is required. In the last decade, we have investigated several different types of organokines in MHO patients, which we refer to as NDO, because of the conflicting definition of MHO. We suggested that complex examination of organokines may add further valuable information about the use of organokines as prognostic factors in morbid obese patients without carbohydrate disturbances. Furthermore, our data showed that organokines may serve as biomarkers in the regulation of lipid and glucose homeostasis in NDO subjects. It can also be established that no degree of obesity is healthy, because despite normal carbohydrate and lipid levels, significant changes can be revealed in the network of organokines. It is important to emphasize that the onset of type 2 diabetes appears to be correlated with the presence of family history. The real question that should be answered is why, with the same excess weight, some individuals develop a number of metabolic alterations (metabolically unhealthy obese, MUO), while other individuals have no metabolic alterations (MHO). Alterations in the network of organokines may at least partly answer this question; however, further studies are needed to clarify this hypothesis.

The use of organokines for therapeutic purposes in obesity is also a subject of research; however, only preliminary data are available in this field. For instance, Matsui et al. [59] previously presented that long-term administration of recombinant PEDF significantly improved body weight, metabolic parameters, and insulin resistance in OLEFT rats. Another in vivo study showed that the administration of PEDF peptides promoted photoreceptor survival in mouse retina models [207] indicating PEDF as a potential therapeutic approach for photoreceptor protection in diabetic retinopathy. However, human application of PEDF requires further research. In addition, due to the favorable effects of FGF21, subcutaneous administration of FGF21 analogues may also become a potential therapeutic goals to reduce obesity- and diabetes-related complications (i) including reducing BMI via inhibition of food intake and increasing energy expenditure; (ii) normalizing glucose, reducing triglyceride and increasing HDL-C levels in the circulation; and (iii) increasing the levels of adiponectin and reducing levels of fibrosis markers [208,209,210]. The safety of once-weekly injected FGF21 analogue PF-05231023 has been investigated both in humans and in non-human primates [209], and the usefulness of pegbelfermin in obese and diabetic patients with nonalcoholic steatohepatitis has also been tested [210]; however, further studies are needed to verify the exact effects of FGF21 in obesity and related metabolic diseases.

## 6. Conclusions

Our increasing knowledge about the diverse roles of organokines in various forms of obesity underline the importance of their further research in large clinical cohorts. Altered pattern of adipokines, hepatokines, myokines and gut hormones might play a crucial regulatory role in metabolic pathways including lipid and carbohydrate metabolism as well as low grade chronic inflammation. We hypothesize that canonical effect of these biologically active peptides and proteins may serve as an intermediate regulatory level that connects the central nervous system and the endocrine, autocrine, and paracrine actions of organs responsible for metabolic and inflammatory processes (Figure 5). The changes in the concentration of organokines are not unidirectional, i.e., the levels of protective organokines decrease and the harmful ones increase during the progress of obesity, but the levels of certain protective organokines may show a compensatory increase, such as in the case of PEDF, PGRN or ghrelin. A better understanding of the function of this delicately tuned network may provide an explanation for the wide range of obesity phenotypes with remarkable inter-individual differences regarding comorbidities and therapeutic responses. Evaluating the effect of anti-obesity agents on organokines may lead to the development of novel personalized therapeutic strategies that may decrease cardiovascular risk and improve the patients’ quality of life and life expectancy.

## Figures and Tables

**Figure 1 metabolites-13-00270-f001:**
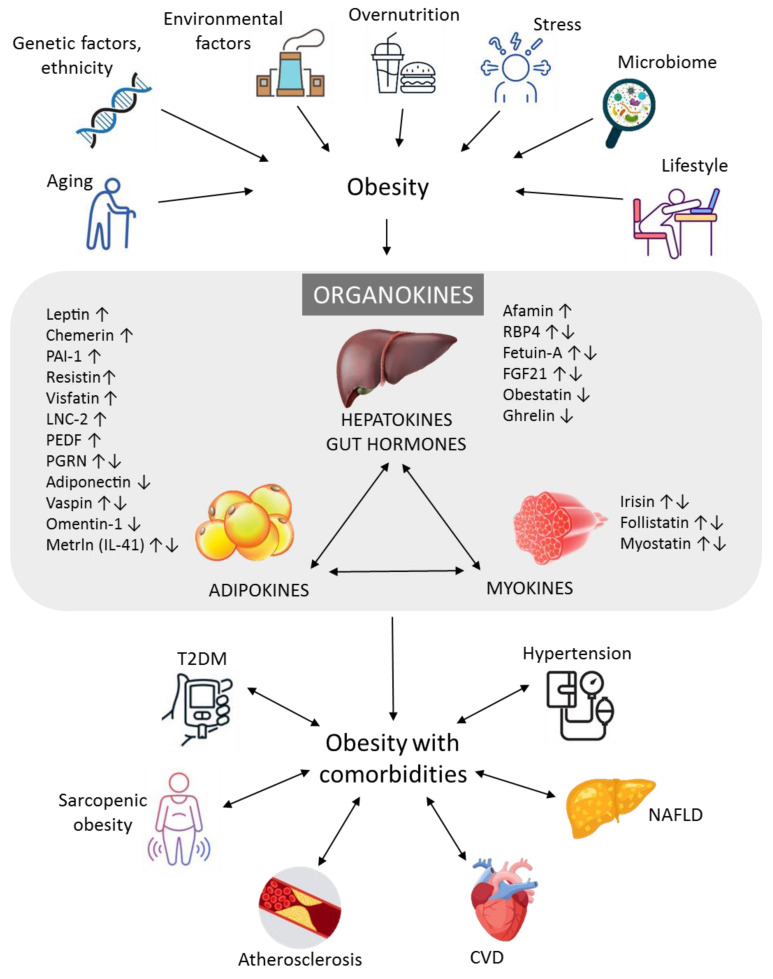
Relationship between different type of organokines in non-diabetic obese (NDO) subjects. Genetic and environmental factors, aging population, stress, sedentary lifestyle, impaired microbiome, overnutrition are associated with growing incidence of obesity. Visceral adipose tissue, liver, gastrointestinal tract and skeletal muscle are potent endocrine organs which regulate energy and metabolic homeostasis via the secretion of adipokines, hepatokines, gut hormones and myokines. These organokines can interact with each other and communicate through various endocrine, paracrine and autocrine pathways. The concentrations of organokines change during the pathological conditions leading to prediabetic and then diabetic status in morbid obesity. The impaired organokine secretion may lead to further unfavourable obesity-associated disorders including sarcopenic obesity, atherosclerosis, CVD and NAFLD. Abbreviations: ↑, increased; ↓, decreased; CVD, cardiovascular disease; FGF21, fibroblast growth factor 21; LCN-2, lipocalin-2; Metrln, meteorin-like, NAFLD; non-alcoholic fatty liver disease; NDO, non-diabetic obese; PAI-1, plasminogen activator inhibitor-1; PEDF, pigment epithelium derived factor, PGRN, progranulin; RBP4, retinol binding protein 4; T2DM, type 2 diabetes mellitus. The graphical components of this figure are available on the website of iStock by Getty Images, Calgary, Alberta, Canada.

**Figure 2 metabolites-13-00270-f002:**
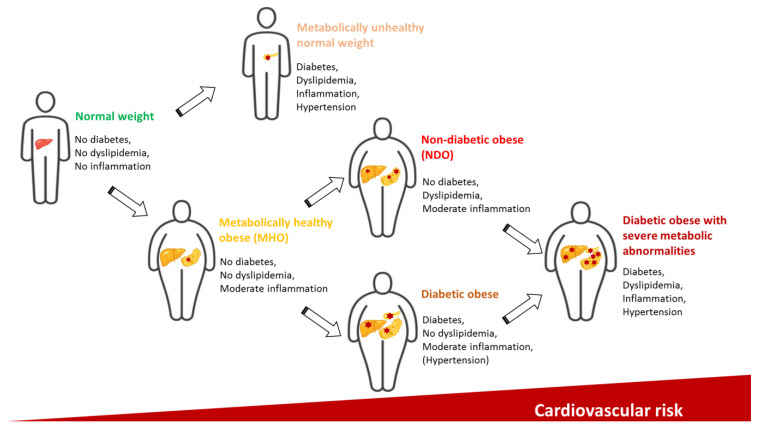
Range of obesity phenotypes with different cardiometabolic status. The graphical components of this figure are available on iStock by Getty Images, Calgary, Alberta, Canada.

**Figure 3 metabolites-13-00270-f003:**
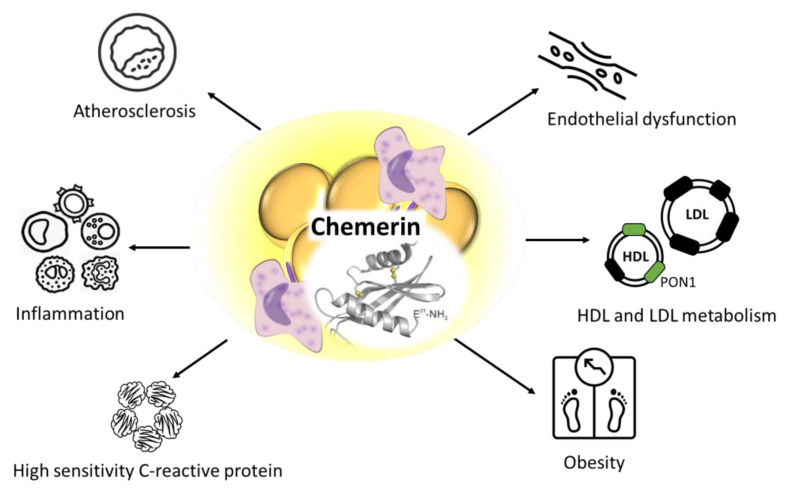
Putative role of chemerin in metabolic processes. The graphical components of this figure are available on iStock by Getty Images, Calgary, Alberta, Canada.

**Figure 4 metabolites-13-00270-f004:**
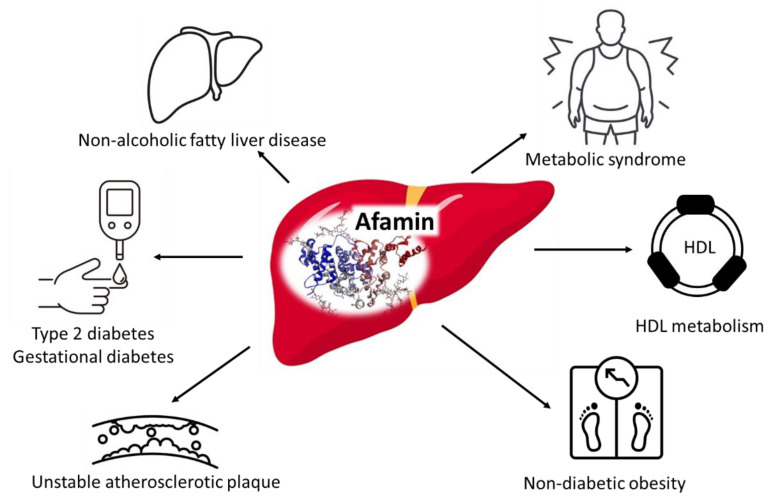
Pathophysiological role of afamin. The graphical components of this figure are available on iStock by Getty Images, Calgary, Alberta, Canada.

**Figure 5 metabolites-13-00270-f005:**
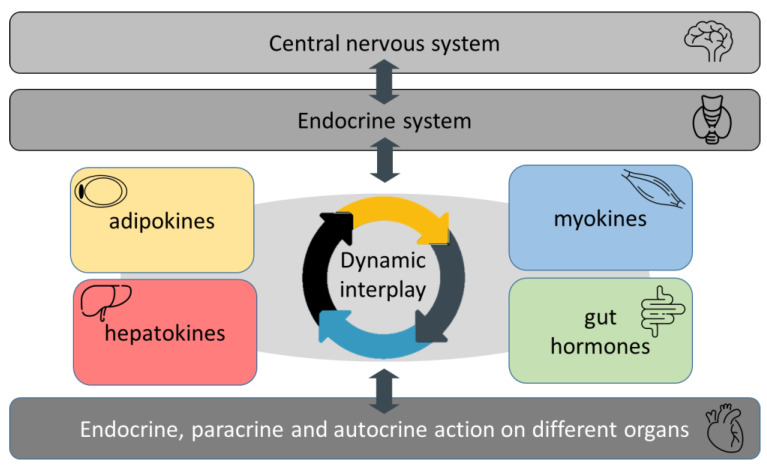
Involvement of organokines in the regulation of neuroendocrine axis. The graphical components of this figure are available on iStock by Getty Images, Calgary, Alberta, Canada.

**Table 1 metabolites-13-00270-t001:** Main characteristics of novel organokines involved in the pathophysiology of obesity.

Organokine	Putative Role in Obesity and Obesity-Related Disturbances	Change	Reference
**Adipokines**			
Chemerin	Mediation of obesity-associated low-grade inflammation; Negative correlation with HDL-linked antioxidant paraoxonase-1 enzyme; Strong correlation with the markers of dyslipidemia including lipoprotein subfractions; Biomarker for obesity and insulin resistance; Mechanistic influence on systolic blood pressure and hypertension.	↑	[23,26,28,32]
PAI-1	Thrombotic effects and inflammation; Biomarker for obesity, insulin resistance and T2DM; Associations with the markers of atherogenic dyslipidemia (correlations with HDL subfractions and ApoAI; PAI-1 release stimulated by small-sized HDL in adipocytes; VLDL was capable of increasing the PAI-1 level in endothelial cells).	↑	[33,35,36,41]
LCN-2	Biomarker for early renal injury; Biomarker for insulin resistance; Biomarker for hypertension; High LCN-2 levels negatively affect muscle physiology.	↑	[45,46,49,51]
PEDF	Anti-angiogenic properties by direct effects on endothelial cells; Role in lipid metabolism by the binding of adipose triglyceride lipase; PEDF expression may be upregulated by a compensatory mechanism; Biomarker for obesity, T2DM and MetS.	↑	[55,56,59]
PGRN	Growth factor in epithelial cells, fibroblasts and adipocytes; Direct binding to TNFR inhibiting neutrophil activation; Contradictory results in the progression of insulin resistance and inflammation; PGRN expression may be upregulated by a compensatory mechanism; Change of PGRN negatively correlated with the improvement of current perception threshold after 6-month alpha-lipoic treatment in T2DM patients with peripheral neuropathy.	↑↓	[60,61,63,65,68]
Vaspin	Insulin-sensitizing and anti-inflammatory agent via a compensatory mechanism; Involvement in hypertension; Involvement in lipid metabolism; Biomarker for obesity, T2DM and NAFLD.	↑↓	[71,72,75,76]
Omentin-1	Insulin sensitizing and anti-inflammatory agent; Biomarker for MetS, hypertension, T2DM and diabetic complications; Involvement in lipid metabolism	↓	[77,79,80,81]
Metrnl	Involvement in thermogenesis in brown/beige adipocytes; Insulin sensitizing and anti-inflammatory effects; Putative involvement in lipid metabolism.	↑↓	[85,86,89]
**Hepatokines**			
Afamin	Association with adiposity, markers of MetS and NAFLD; Biomarker for the prevalence and incidence of T2DM and gestational diabetes; Biomarker for unstable atherosclerotic plaque; Involvement in HDL metabolism.	↑	[96,97,99,101]
RBP4	Strong correlation with the markers of dyslipidemia including lipoprotein subfractions and ApoAI; Association with adiposity, insulin resistance, MetS and T2DM.	↑↓	[103,105,106,109]
Fetuin-A	Strong correlation with the markers of dyslipidemia including lipoprotein subfractions; Putative association to HDL proteome; Biomarker for obesity, MetS and T2DM; Biomarker for subclinical atherosclerosis.	↑↓	[109,113,115,117,123]
FGF21	Insulin-sensitizing effects; Regulator of thermogenesis and lipolysis in the brown adipose tissue; FGF21 expression may be upregulated by a compensatory mechanism in obesity, T2DM and hypertension; Short term moderately intensive physical activity improved FGF21 levels in T2DM patients with peripheral neuropathy.	↑↓	[127,128,133,134,135]
**Gut hormones**			
Obestatin	Anorexigenic and insulin-sensitizing effects; Involvement in the pathophysiology of obesity, insulin resistance and T2DM; Involvement in the regulation of lipid metabolism including VLDL and HDL subfractions, correlation with mean LDL size.	↓	[139,142,145]
Ghrelin	Orexigenic effects, stimulating gastric motility and hepatic glucose secretion and reducing insulin secretion; Inducing growth hormone and glucocorticoid secretion; Ghrelin expression is markedly decreased in obese individuals independently of gender.	↓	[147,152,203]
**Myokines**			
Irisin	Insulin sensitizing hormone; Involvement in lipid metabolism; Biomarker for MetS, T2DM and cardiovascular diseases; Increased levels after physical activity; Decreased levels after weight loss due to bariatric surgery.	↑↓	[171,178,185,187]
Myostatin	Negative regulator of skeletal muscle growth; Levels are decreasing with age and exercise; Related to systemic inflammation and sarcopenic obesity.	↑↓	[191,193,195]
Follistatin	Promoting muscle mass and function; Potential therapeutic use for the treatment of muscle wasting in cachexic conditions; Involvement in beige and brown adipose tissue differentiation.	↑↓	[197,202]

Abbreviations: ↑ increased; ↓ decreased; ApoAI, apolipoprotein AI; FGF21, Fibroblast growth factor 21; HDL, high-density lipoprotein; LCN-2, lipocalin-2; LDL, low-density lipoprotein; Metrnl, Meteorin-like; MetS, metabolic syndrome; NAFLD, non-alcoholic fatty liver disease; PAI-1, plasminogen-activator inhibitor-1; PEDF, pigment epithelium derived factor; PGRN, progranulin; RBP4, Retinol-binding protein 4; TNFR, tumor necrosis factor alpha receptor; T2DM, type 2 diabetes mellitus; VLDL, very low-density lipoprotein.

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
