# Peer review of "Crucial Regulatory Role of Organokines in Relation to Metabolic Changes in Non-Diabetic Obesity"

_metabolites, 2023, doi:10.3390/metabo13020270_

Round 1
Reviewer 1 Report
The authors of the review "Crucial regulatory role of organokines in non-diabetic obesity" provide an extensive, up-to-date and useful summary on new organokines (adipokines, hepatokines and myokines) and their correlation with obesity.
I have some concerns regarding the setting of the introduction and the first part of paragraph 5 "Treatment and future directions." In the title, the introduction and the first two figures, it seems that obesity necessarily constitutes the first step in the development of prediabetes and, subsequently, diabetes. This pathway would be due to alteration of different organokines. Any other metabolic alteration would be secondary to this process. In clinical reality, things are not exactly like that. Many individuals with dyslipidemia, hypertension, NALF, atherosclerosis, hyperinsulinism, and hyperuricemia have perfectly controlled blood sugar. The onset of type 2 diabetes appears to be strongly correlated with the presence of family history. The real question that should be answered is why, with the same excess weight, some individuals develop a number of metabolic alterations (metabolically unhealthy obese, MUO), while other individuals have no metabolic alterations ( metabolically healthy obese, MHO). Could the prevalence of some organochines over others provide an answer to this question? Could genetic makeup help clarify the picture as for example is suggested for PAI-1?
Based on these considerations, what is reported in Figure 2, from which there seems to be a necessary transition from MHO to obese nondiabetic (NDO) to MUO, does not seem convincing to me. In my opinion, the title of this review could be similar to "An overview of new organokines in relation to metabolic changes in obese patients," without necessarily mentioning the questionable category of metabolically healthy obese individuals (MHO).
The review never mentions hypertension, which is certainly one of the most important risk factors for the development of cardiovascular disease. A correlation with the development of hypertension is demonstrated for the best-known organochines. What are the new organokine data on this topic? I think this is an important point to clarify and develop in this review.
In general, the relationship between obesity and organokines is relatively simple: organokines that have a protective action on the cardiovascular system decrease as body weight increases, whereas organokines that stimulate negative metabolic/inflammatory pictures increase as weight excess increases. If we think in finalistic terms, this picture is believable. However, it is possible to hypothesize that protective mechanisms may be established in the case of weight excess. For PEDF, this possibility has been hypothesized. This hypothesis would make the organokines network more complex but also more fascinating and could perhaps partly explain what motivates the existence of the MHO phenotype.
It would be useful to include a summary table of the effects of various organokines (old and new) in accordance with the presence of obesity: which ones increase and which ones decrease, on which metabolisms they act, what evidence of association they have with cardiovascular risk factors.
Reviewer 2 Report
Manuscript ID: metabolites-2141826Crucial regulatory role of organokines in non-diabetic obesity
Some sentences seem ambiguous, correct them.
I have reviewed the aforementioned manuscript, below are my suggestions for authors.
Dear authors,
Rewriting the contents coherently will help publish this manuscript.
1. Correct the format errors
2. Check this sentence "Omentin-1 increased 223 Akt phosphorylation in the insulin signaling pathway in the
absence and presence of insulin"
3. Include studies about ghrelin and vaspin associated with obesity.
4. I suggest the authors to give graphical representation data.
5. How does the probability of obesity.
6. Include the two variants of Follistatin.
7. Focus on clear presentation of data.
8. Use italics wherever necessary.
9. Fig 3: check the caption. Moreover, modify the figure effectively otherwise consider removing from the article since it seems less significant.
10. No need to illustrate "conclusion" instead concentrate on briefly summarizing your viewpoints on the results of your study.
11. Missing words in some places make the sentences incomplete, correct them.
12. Try to demonstrate adipokines using pictorial representation wherever possible.
13. Paragraphs are disproportionate, modify properly.
14. How do you think environmental factors contribute to obesity?
15. Is it necessary to write organokines, adipokines, hepatokines, myokines in italics?
16. What is the major differentiation factor between MHO
and NDO?
17. Among the factors, what about ethnicity?
18. Manuscript needs language correction, kindly follow that.
Reviewer 3 Report
Thank you for the opportunity to review this paper.
This is a useful review article summarizing literature that describes the role of organokines. Specifically, the authors focus on the latest finding on the role of adipokines, hepatokines, and myokines in obesity and regulation of lipid and glucose metabolism. A brief overview of their relationship with an increased/decreased risk of diabetes and cardiovascular diseases. As a novelty, the authors chose to discuss all findings on novel organokines in non-diabetic obesity, which is a transient state between metabolically healthy obesity and metabolically unhealthy obesity.
The manuscript is very well written and the objective of the review is clearly stated. The references are up to date and appropriate.
I have only a minor suggestion. In the abstract and in the Introduction section (page 3, line 84) the authors state that their review is focusing on “severe obese non-diabetic subgroup”. This is unclear in the review as no “severe” group of patients is discussed, but only obesity without diabetes irrespective of the grade of obesity.
Round 2
Reviewer 1 Report
The manuscript 'Crucial regulatory role of organokines in relation to metabolic changes in non-diabetic obesity' is certainly much improved in its current version and definitely deserves to be published because it provides a comprehensive and current view of the role of organokines in obesity. The new figures and the summary table are very useful. I however still have doubts about the first two figures where, despite the correct clarifications added to the text, the development of prediabetes and then diabetes seem to be necessary steps for the other metabolic alterations onset and in which hypertension still does not appear. I would suggest changing Figure 1 as shown below. On the other hand, I would not know how to change Figure 2, but it still seems to me that it could create misunderstandings.
Please look in the attached file to see the changes I suggest for Figure 1

Reviewer 2 Report
Dear authors,
Though this manuscript has been revised by the authors, still there some grammatical errors, kindly correct them.
Other comments are as follows:
L.No: 34- are you sure? Carbohydrate- rich nutrition? Reconsider this,think about carbohydrate-rich food.
L.No: 61 - rewrite the sentence.
L.No: 82 to 84 - correct those sentences.
L.No: 126 to 127 - correct the sentences.
L.No: 130 - Either COVID - 19 or SARS-CoV-2, check this out.
L.No: 121 to 124 - Correct the sentences and make it simple.
L.No: 155 to 158 - rewrite the sentences, clearly.
Fig: 3 - HDL and metabolism?
Check if you have precisely placed the images
Differentiate c-reactive protein
Caption and figure should be corrected.
L. No: 164 to 190 - Correct the sentences
Fig. 4 - Increase the quality of the image. Caption should have "Afamin"
L.No: 418- In vitro must be in italics.
Culminate your review of each adiponectin lucidly and avoid disarray within paragraphs.
Correct the title of Table 1.
Fig.5: Caption didn't match with the figure
Instigate the molecule in the context aptly before writing about the research data or outcomes of the previous researches.
Please consider correcting the grammatical errors once again in this manuscript.
Round 3
Reviewer 2 Report
Dear authors,
You have followed the suggestions.
I wish you success in publishing your manuscript.